# Organomorphic Silicon Carbide Reinforcing Preform Formation Mechanism

Evgeny Bogachev 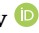

JSC Kompozit, 4 Pionerskaya, Korolev, 141070 Moscow, Russia; eug-bogatchev@mail.ru; Tel.: +7-495-513-23-06

**Abstract:** Development of the organomorphic ceramic-matrix composites (CMCs), where the reinforcing preform is built using polymer fibers subject essentially to hot pressing, was motivated by a desire to obtain much higher structural uniformity as well as to reduce the number of the process steps involved in the production of CMCs. This paper addresses the peculiarities of the organomorphic silicon carbide preform formation process. Using X-ray phase analysis, tomography, mass and IR spectroscopy, and thermomechanical and X-ray microanalysis, both the properties of the initial fibers of polycarbosilane (PCS)—the silicon carbide fiber precursor—and their transformation in the preform while heated to 1250 °C under constant pressing at 10–100 kPa were studied. Analysis of the data obtained showed the organomorphic SiC preform relative density at a level of 0.3–0.4 to be ensured by self-bonding of the silicon carbide preform, resulting from the fact that during the low-temperature part of pyrolysis, easily polymerizing substances are released leaving a high coke residue, thus cementing the preform. Another possible factor of SiC framework self-bonding is the destruction of the polymer fibers during pyrolysis of various PCS preforms differing in their methylsilane composition (for example, dimethylsilane), where deposition of silicon carbide on the contacting fibers starts as early as at 450–500 °C.

**Keywords:** polycarbosilane (PCS); fiber; pyrolysis; self-bonding; silicon carbide; organomorphic preform (OP)

## 1. Introduction

At the core of ceramic matrix composites (CMCs) is their reinforcing system. The existing technology for fabricating the reinforcing system involves multiple stages. These stages include a number of operations, namely, the production of elementary organic fibers from the relevant precursor, whereupon they are subject to step-by-step pyrolysis as part of a multifilament (made of thousands of filaments) tow: manufacturing of a fabric or a tape to be impregnated with a binder to obtain the prepreg; molding of the preform by curing; and carbonization that involves high-temperature annealing. There are certainly other ways for obtaining the reinforcing system using the available mechanical binding methods (multi-dimensional weaving, needle punching, and braiding); however, the reinforcing frames obtained using the aforesaid methods show a relatively low density and cannot be used as a basis for structural composites without the use of a binder. The numerosity of the operational processes involved results in the high fabrication cost of the preform as well as the CMC itself.

A special feature that defines the CMCs obtained using conventional methods is their rather non-uniform structure that, for the time being, cannot match the structural uniformity of graphite, metal, and ceramics. This prevents us from expanding the scope of CMC application as an alternative to these materials. It seems that any efforts to solve this problem by modifying a conventional preform, one way or another, are bound to fail. The use of multifilament tows, which are immanent features of the state-of-the-art technology, will result intrinsically in a non-uniform distribution of the reinforcement in relation to the composite volume, since the filament diameter range between 7 μm and 15 μm, and the size

of the voids between the multifilament tows will make up 300–400 μm to 700–1000 μm and between the filaments—0.3–0.7 μm. The notable micro non-uniformity of the reinforcing preform will impose a natural limitation on its minimum thickness. Obtaining a composite material where the elementary fibers alternate uniformly with the matrix, thus bringing the dimensional limit of the material essentially about the diameter of the fiber itself, would bring the lower limit of the thickness of the parts down to 150–200 μm. Such a possibility, along with a notable reduction in the number of the process steps, may expand significantly the scope of application of CMCs through inclusion of many promising applications where CMC can be used instead of graphite, traditional ceramics, or metal.

The search for a solution of this problem resulted in the development of the so-called organomorphic CMCs (C/C, C/SiC, and SiC/SiC) reinforced with a framework based on organic polymer fibers used for the most common carbon and ceramic fibers, namely, polyacrylonitrile (PAN), polycarbosilane (PCS), and polysilazane (PSZ) [1].

Using PAN fibers as an example, hot pressing (up to 1000 °C) of PAN preforms was found [2] to provide carbon reinforcing frames, showing a high relative density of up to 0.4 and an open porosity of 50–60% represented by pores of an equivalent diameter ranging from several micrometers to several tens of micrometers. The method used to build the reinforcing carbon preform was found to provide self-bonding (as inherited after pyrolysis) of the filaments. The new quality of the carbon preforms made it possible both to provide an effective substitute for the molybdenum electrodes used for ion thruster accelerating electrodes [3] and to find more effective methods to CVI them with the silicon carbide matrix [4].

Self-bonding during so-called hot pressing (up to 170 °C) is a well-known phenomenon typical of natural fibers such as cotton and wood [5–8]. This phenomenon, resulting from a physical contact between the fibers as well as the presence of a bonding-conducive polymer on their surface, makes it possible to fabricate high-strength boards made of natural materials without any binder.

However, the organomorphic CMC preforms made of polymer fibers are subject to annealing at up to 1800 °C, and they are not limited by the process temperatures characteristic of the natural fibers. Therefore, it is important to determine the presence of self-bonding in the PCS preforms pyrolized under pressing, including at the low-temperature (up to 400 °C) pyrolysis stage. Formerly, self-bonding was observed in pyrolized PAN-based preforms [1,2].

The PAN- and PCS-fibers differ significantly in their chemical composition and structure. At the same time, it seems that the self-bonding discovered previously for PAN fibers during pyrolysis under pressure must be basically of general nature, and the governing laws of obtaining the carbon and SiC preforms, respectively, must be more or less the same.

## 2. Materials and Methods

The PCS fibers of Kompozit JSC make, thermally stabilized at 200 °C (molecular mass distribution $M_n \geq 1600$, polydispersity coefficient $D \leq 1.8$, 90% wt. residue after etching treatment) in air, 19–24 μm in diameter, were studied with the Skyscan-2011 nanotomograph (London, United Kingdom) using the following scanning parameters: U = 50 kV, I = 200 mkA, rotation angle—0.3° as averaged over 5 frames, and spatial resolution—0.3 μm/pixel.

The PCS fiber behavior, while under load, was studied with the thermomechanical analysis method (TMA) using the TMA Q400 EM (New Castle, Delaware, USA). The test sample in the form of layers of fibers, 7 mm in diameter and 3.4 mm in thickness, was placed in a special pot made of high-density organomorphic C/C, 7 mm in diameter, 11 mm in height, and 1.5 mm in wall thickness. Next, the fiber sample was pressed with a quartz indenter, 2.54 mm in diameter (Figure 1), to be heated up to 400 °C in air at a rate of 1 deg/min, under a static load of 0.588 N and a dynamic load of (±0.294 N), keeping a record of the time change in the elasticity modulus and sample thickness.

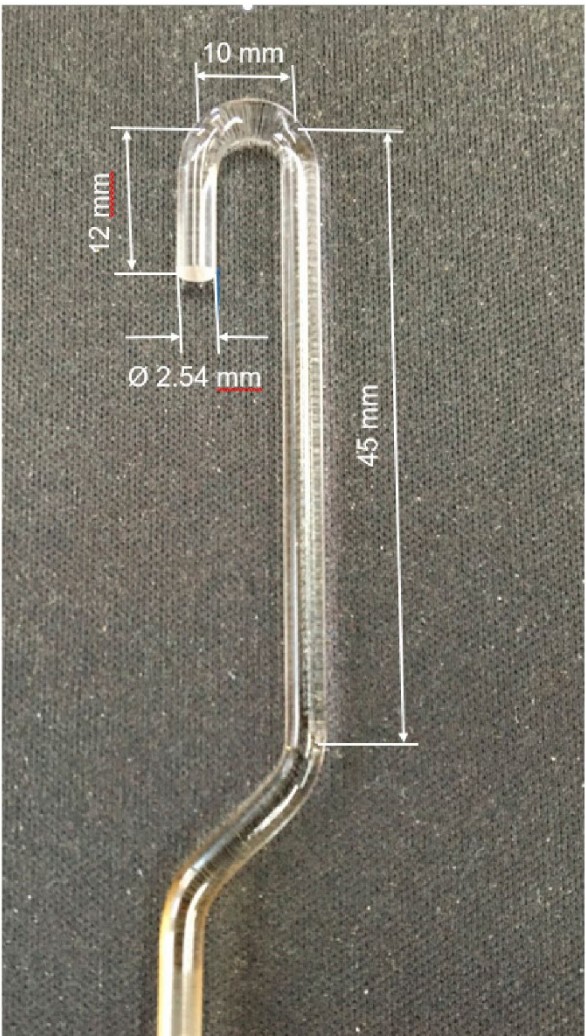

**Figure 1.** EXPANSION probe.

The structure and functional groups of the PCS fiber macromolecules, both before and after TMA, were analyzed using X-ray tomography with the XT H 320 LC X-ray tomography system (METRIS, Tring, UK) and using IR spectroscopy with the Netzsch STA 449 F3 Jupiter (Selb, Germany).

The PCS fiber thermal decomposition kinetics was studied in high-purity nitrogen flow with the TGA-DSC method using the TA Instruments SDT 650 synchronous differential thermal analyzer (New Castle, DE, USA) at the rate of a temperature rise of 10 degrees per minute.

The PCS fiber gas emission during heating up to 800 °C in vacuum was studied using the laboratory mass spectrometer equipped with the CIS300 quadrupole mass analyzer (Sunnyvale, CA, USA). The PCS fibers, 1–5 mg in weight, were placed in a quartz ampoule connected to the quadrupole mass spectrometer vacuum system. The ampoule was pumped out at the room temperature down to ~$10^{-3}$ Pa and connected to the mass spectrometer chamber. While keeping continuous record of the mass spectra, the ampoule was subject to heating from the room temperature up to 800 °C at a constant rate of 5 deg/min. The mass spectra were recorded every 90 s, over the mass number range of 1–220. Simultaneously, record of the pressure in the mass spectrometer chamber (sensor MID-2) and in front of the mercury diffusion pump behind the nitrogen trap (sensor MID-1) was kept.

To obtain the organomorphic preforms from the silicon carbide fiber, the fibers in the form of a tape composed of 150-filament threads were placed in a graphite container with the internal dimensions of 120 × 60 × 28 mm in the directions of 0° × 0° and 0° × 90°,

whereupon a commensurable massive graphite punch was installed onto the fibrous PCS workpiece, placing an additional load on the top [9]. The total load on the fibrous polymer workpiece (cover + load) was 700 N. The ready assembly was installed in a vacuum resistance furnace to be heated up to 1250 °C in non-oxidizing gas (nitrogen) at a rate of 300–400 deg/h. After cooling down, the organomorphic SiC preform was removed from the container to determine the density, whereupon test samples of about 20 × 10 mm in size were cut out of it with a sharp knife for microstructural analysis.

The density and porosity of SiC preforms were measured using the Archimedes method.

To study the structural changes in the material, the scanning electron microscope FEI Quanta 600 FEG (Eindhoven, The Netherlands) with field emission and integrated microanalysis system EDAX TRIDENT XM 4 was used. For X-ray phase analysis, the Empyrean diffractometer (PANalytical B.V., Willmington, Delawer, USA) equipped with the specialized HighScore Plus software for phase analysis with a built-in database of reference structures PAN-ICSD (Inorganic Crystal Structure Database) was used.

## 3. Results

### 3.1. Initial PCS Fiber Properties

Analysis of the initial PCS fibers reveals a completely amorphous and disordered nature of the structure, as evidenced by the fact that the small-angle scattering shows no reflection (Figure 2).

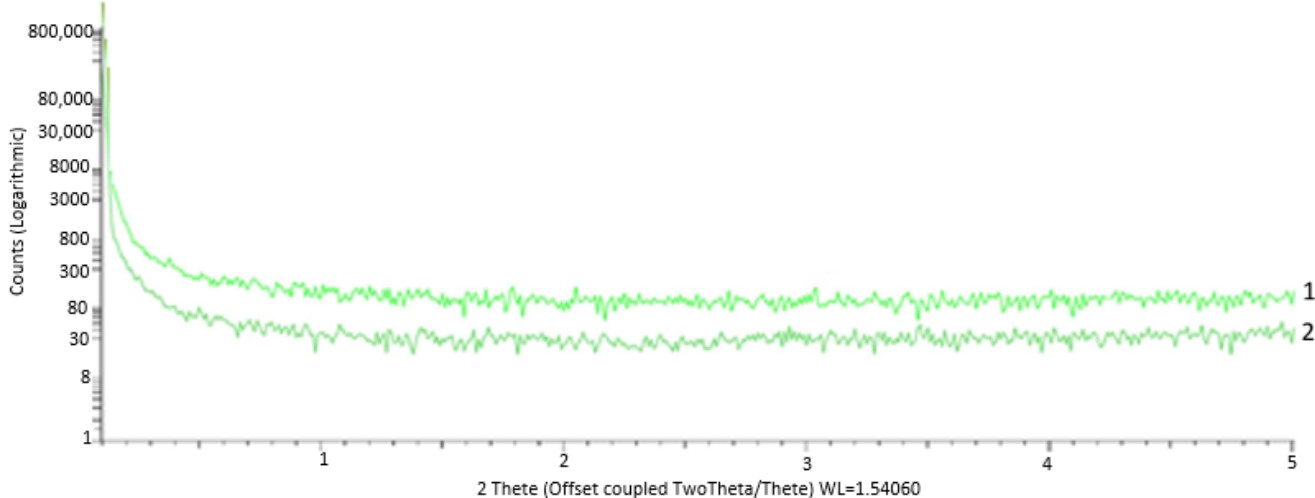

**Figure 2.** Meridional (1) and equatorial (2) scans of PCS fiber bundles in the small-angle area.

Microtomography of the polymer fibers shows no systematic difference in the near-surface and intra-volume areas (Figure 3).

It is clear, however, that there are more or less dense microvolumes in the fibers, which speaks for certain structural defects of the organic filaments.

### 3.2. Analysis of the PCS-Fiber Properties during the Pyrolysis

Thermomechanical analysis of the PCS fiber bundle 3-mm thick reveals complex deformation of the test sample under simultaneous static/dynamic loading with rising temperatures (Figure 4).

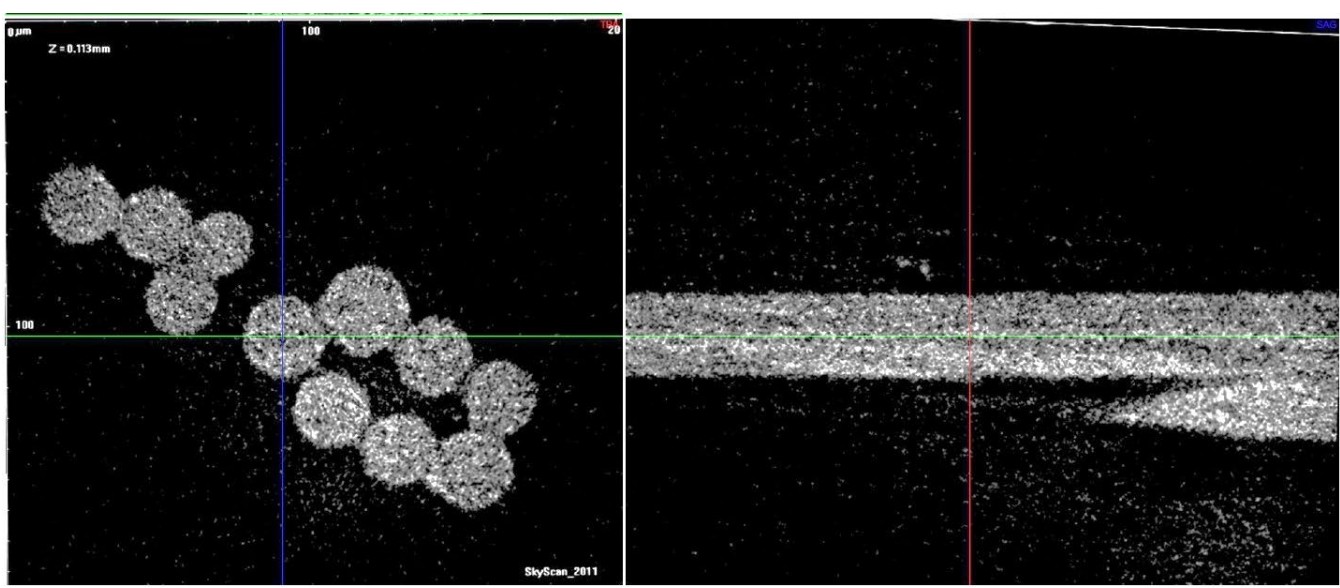

**Figure 3.** Interperpendicular sections of PCS fibers.

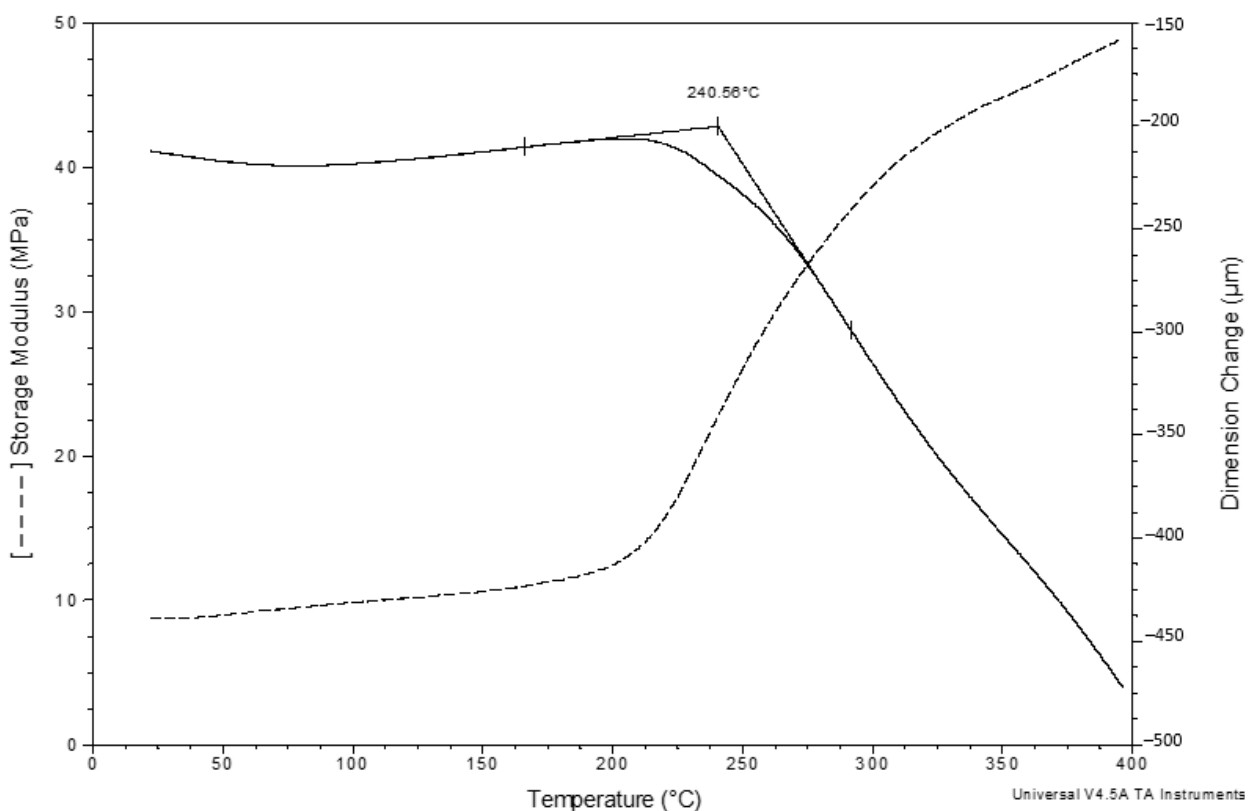

**Figure 4.** Elasticity modulus vs. temperature curves obtained for the PCS fiber bundle.

What draws attention is the timely coincidence of the sharp decrease in the sample thickness and the rapid increase in the elasticity modulus—both processes start at 215–220 °C. The decrease in thickness can be explained both by the partial tangential displacement of the fibers towards the pot walls with rising temperature due to friction reducing between the fibers and the increase in the degree of their contact simultaneously with the deformation of the fibers themselves at a temperature preceding the pyrolysis onset temperature. On the DTG curve, the onset of gas emission from PCS fibers practically corresponds to the start of the events during TMA (Figure 5).

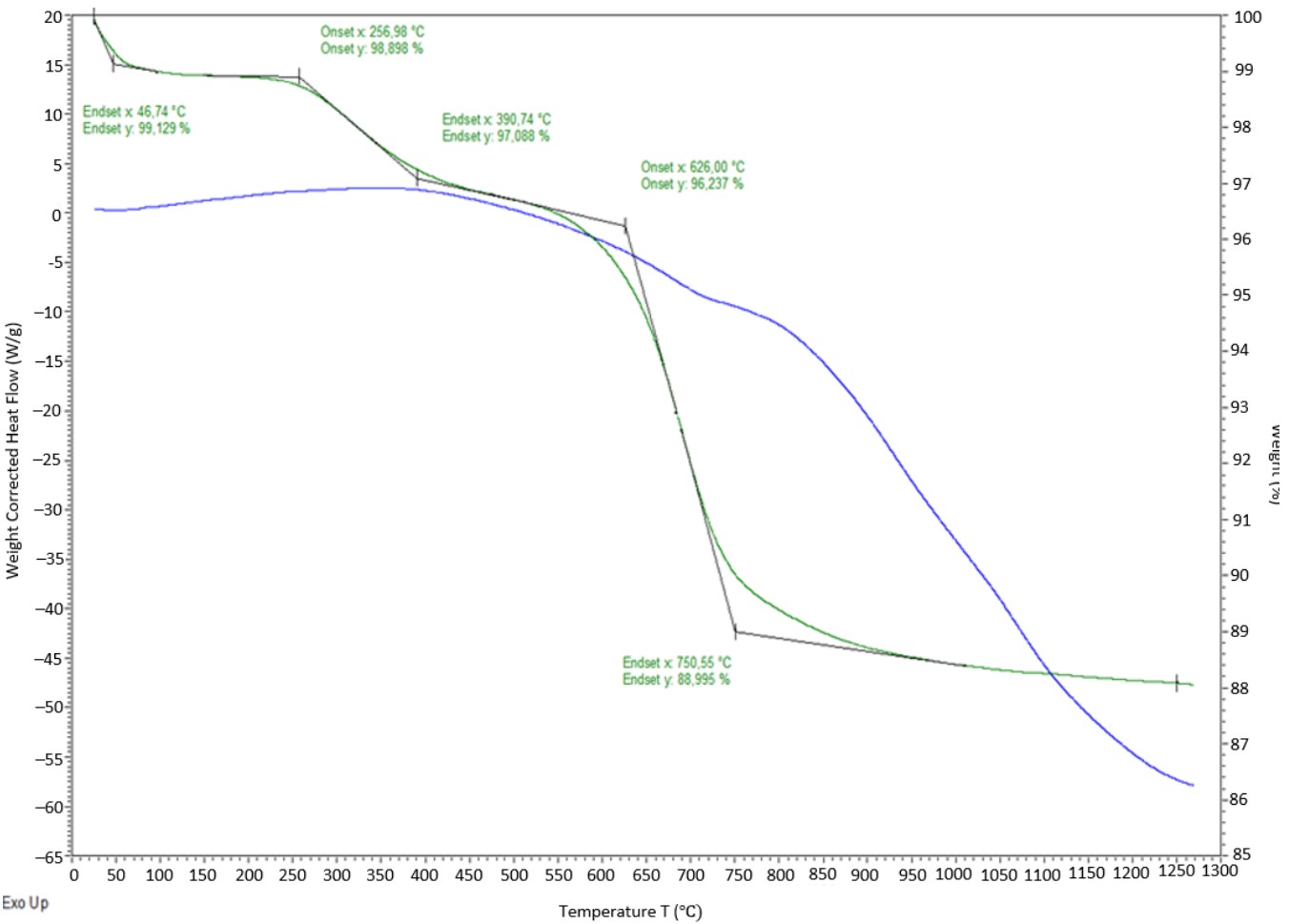

**Figure 5.** Thermogravimetric analysis of PCS fibers.

Mass spectroscopy of the PCS fibers shows that, according to the MID-2 pressure sensor readings, the low-temperature peak of gas emission starts at 215 °C, reaches its maximum at 270 °C, and ends at 470 °C (Figure 6).

The MID-1 pressure sensor fails to record this broad peak, meaning that in the specified temperature range, release of volatile substances with the mass-to-charge ratios of 18 ($H_2O$), 28 (CO), and 44 ($CO_2$) that condense at the liquid nitrogen temperature takes place. These low-molecular weight compounds ensure the loss of about 2% wt. of PCS fibers in the temperature range under study, i.e., 215–470 °C (Figure 5). Removal of oxygen-containing compounds from the composition of the fibers results inevitably in the partial breakage of the oxygen cross-link in PCS macromolecules, ensuring an increase in their mobility and ability for autohesive interaction.

The second (higher) peak of gas emission starts at 500 °C, reaches its maximum at 660 °C, and ends at 740 °C. In this case, the MID-1 pressure sensor does show it, meaning that it is associated mainly with release of hydrogen, since it is not frozen at the liquid nitrogen temperature. The mass spectrometry results also correlate with the DTA data of differential thermal analysis (Figure 5). However, according to [10], a significant mass loss (more than 7% wt.) is provided by the evolution of both hydrogen and methane.

Thus, in the range of 215–400 °C, transformations occur both in the composition and in the structure of the PCS fibers, leading both to a decrease in the thickness of the compressed sample and an increase in its elasticity modulus.

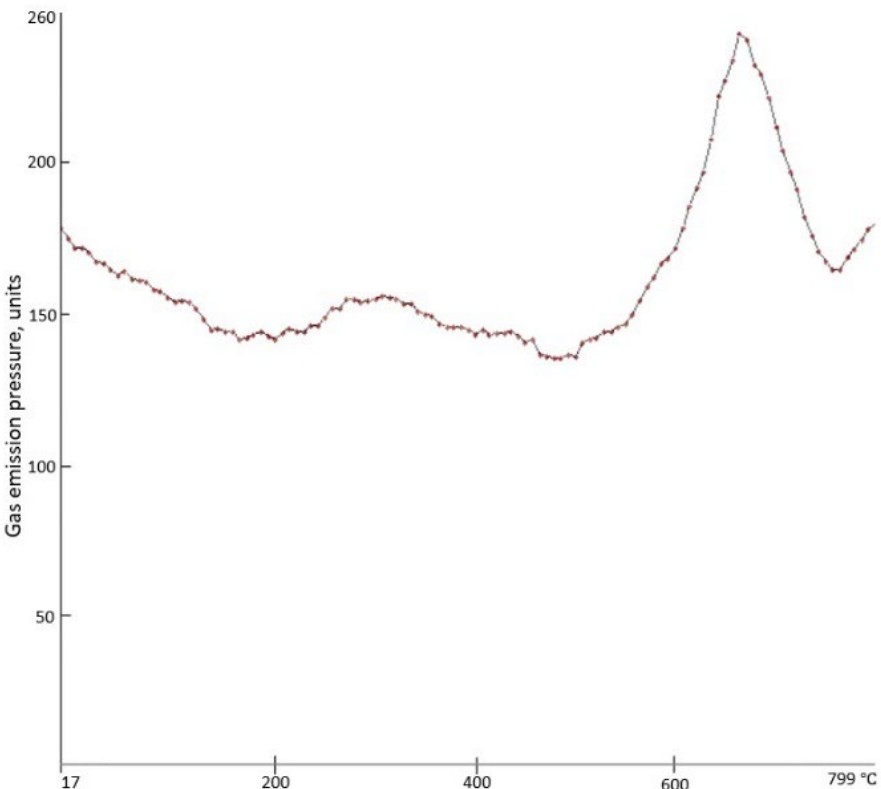

**Figure 6.** Changes in the total gas release from the PCS fibers when heated in the mass spectrometer.

*3.3. Analysis of the PCS-Based Organomorphic Preform (OP) Microstructure and Properties after Different Stages of the Pyrolysis*

After TMA, the test sample represents a fairly dense tablet (Figure 7).

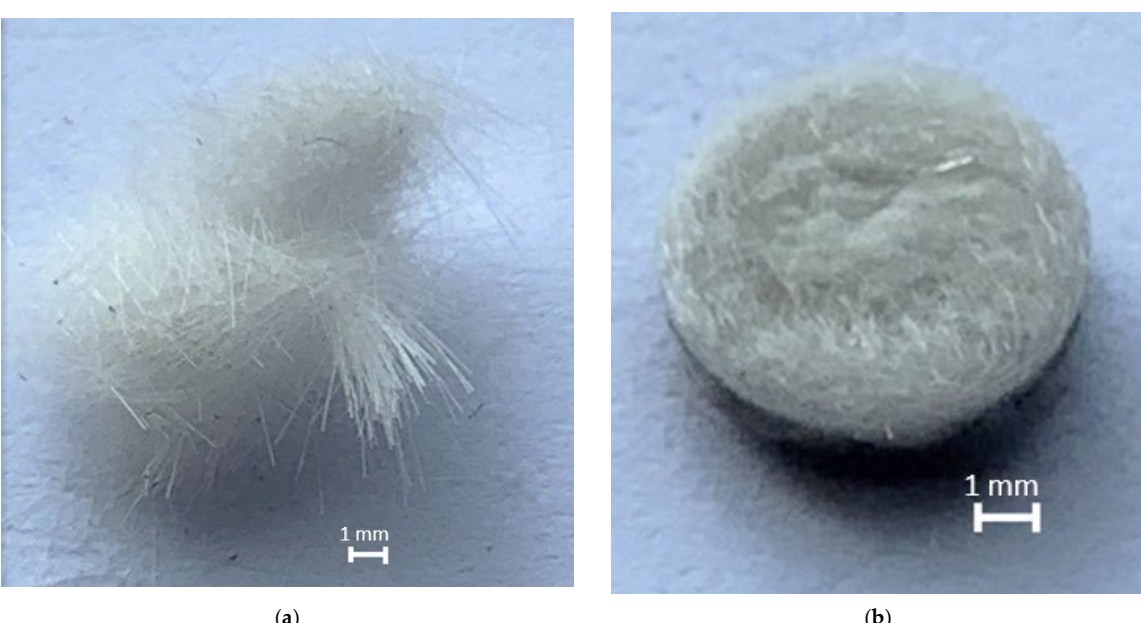

(**a**)                                           (**b**)

**Figure 7.** PCS test sample before (**a**) and after (**b**) thermomechanical analysis at up to 400 °C.

Analysis of the PCS sample microstructure after TMA shows clearly visible insular buildups on the fiber surface (Figure 8), while in the initial state the fiber surface was free of them.

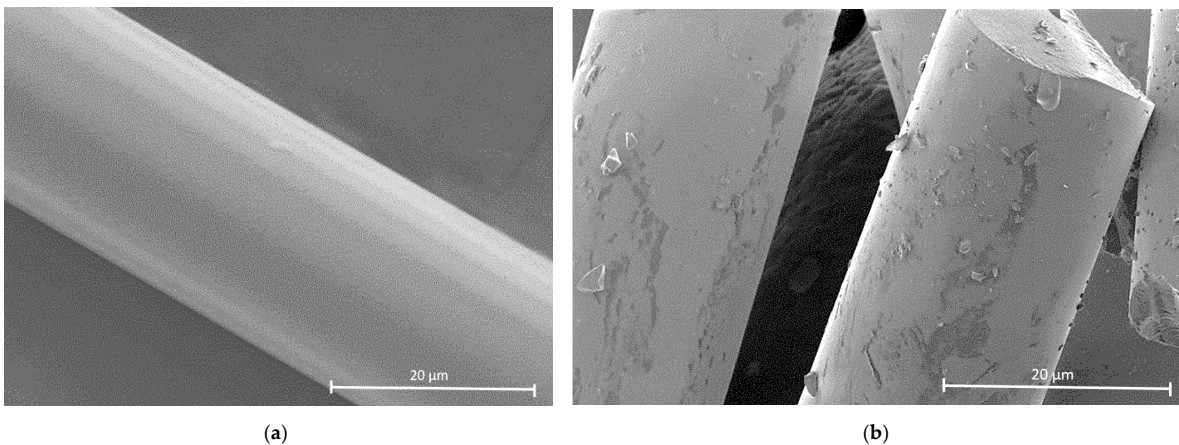

(**a**)　　　　　　　　　　　　　　　　　(**b**)

**Figure 8.** Morphology of PCS fibers before (**a**) and after (**b**) thermomechanical analysis at up to 400 °C.

These buildups are not present on all fibers and are characterized by irregular concentrations. It is probably for that reason that, after TMA, the PCS sample shows less irreversible compression in the indenter pressure area than the PAN sample, the entire surface of which is covered with resin-forming compounds that effectively bind the filaments [2] (Figures 9 and 10).

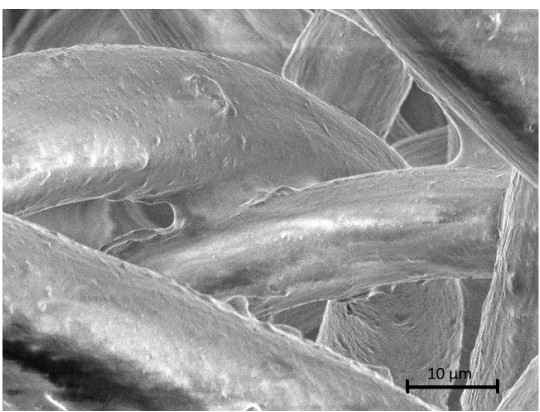

**Figure 9.** Morphology of PAN fibers after pressing at 180 °C [2].

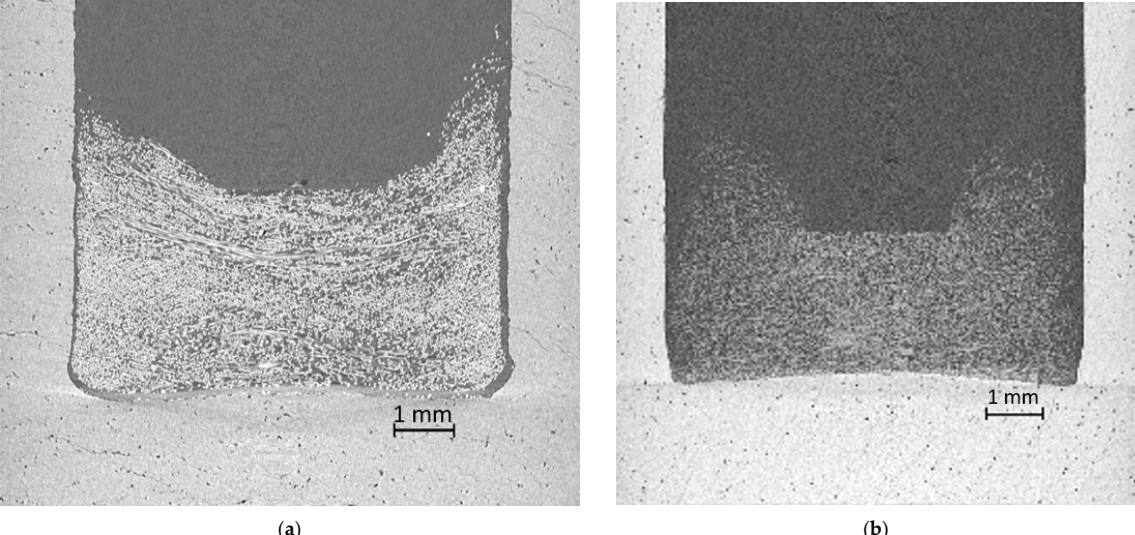

(**a**)　　　　　　　　　　　　　　　　　(**b**)

**Figure 10.** Tomographic images after TMA: (**a**) PCS fiber and (**b**) PAN fiber [2].

Analysis of the PCS fiber sample composition using IR spectroscopy showed the sample underwent significant changes during TMA; what remains in the IR absorption spectrum of the fiber after TMA are signals at 780, 1220 cm$^{-1}$ (Si–CH$_3$ bonds), 1440 cm$^{-1}$ (C–H bonds), and 980 cm$^{-1}$ (Si-CH$_2$–Si bonds) [10,11] (Figure 11).

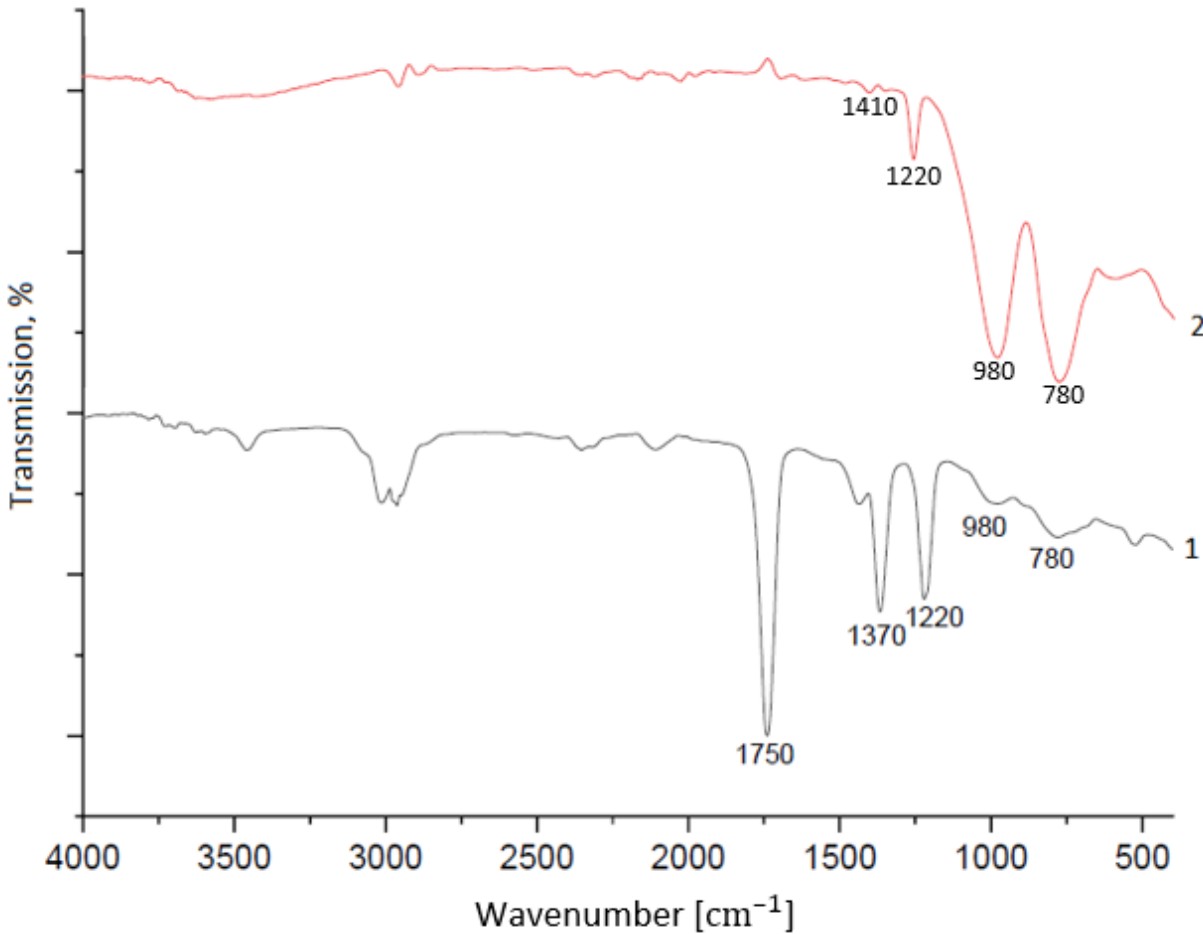

**Figure 11.** Results of IR-spectroscopy of the PCS fibers before (1) and after (2) TMA.

After TMA, signals at 1370, 1750, and 2960–3020 cm$^{-1}$, part of which are related to oxygen-containing bonds, disappear. These changes also correlate with both the results of mass spectroscopy and the compressed fiber transformation. The persistence of the irreversible deformation of the PCS sample after TMA, albeit to a lesser extent than in the case of the PAN sample (Figure 10), explains the increase in the elastic modulus of the compressed region during TMA; formations protrude onto the surface of the filaments and interact with each other autohesively to fix the gradual transformation of individual fibers into a single whole.

Therefore, the insular nature of the self-bonding areas of the fibers does not prevent the formation of dense and highly processable reinforcing silicon carbide frames of various reinforcement structures that proceed from organic to inorganic state, accompanied by their shrinkage as a whole (Figure 12).

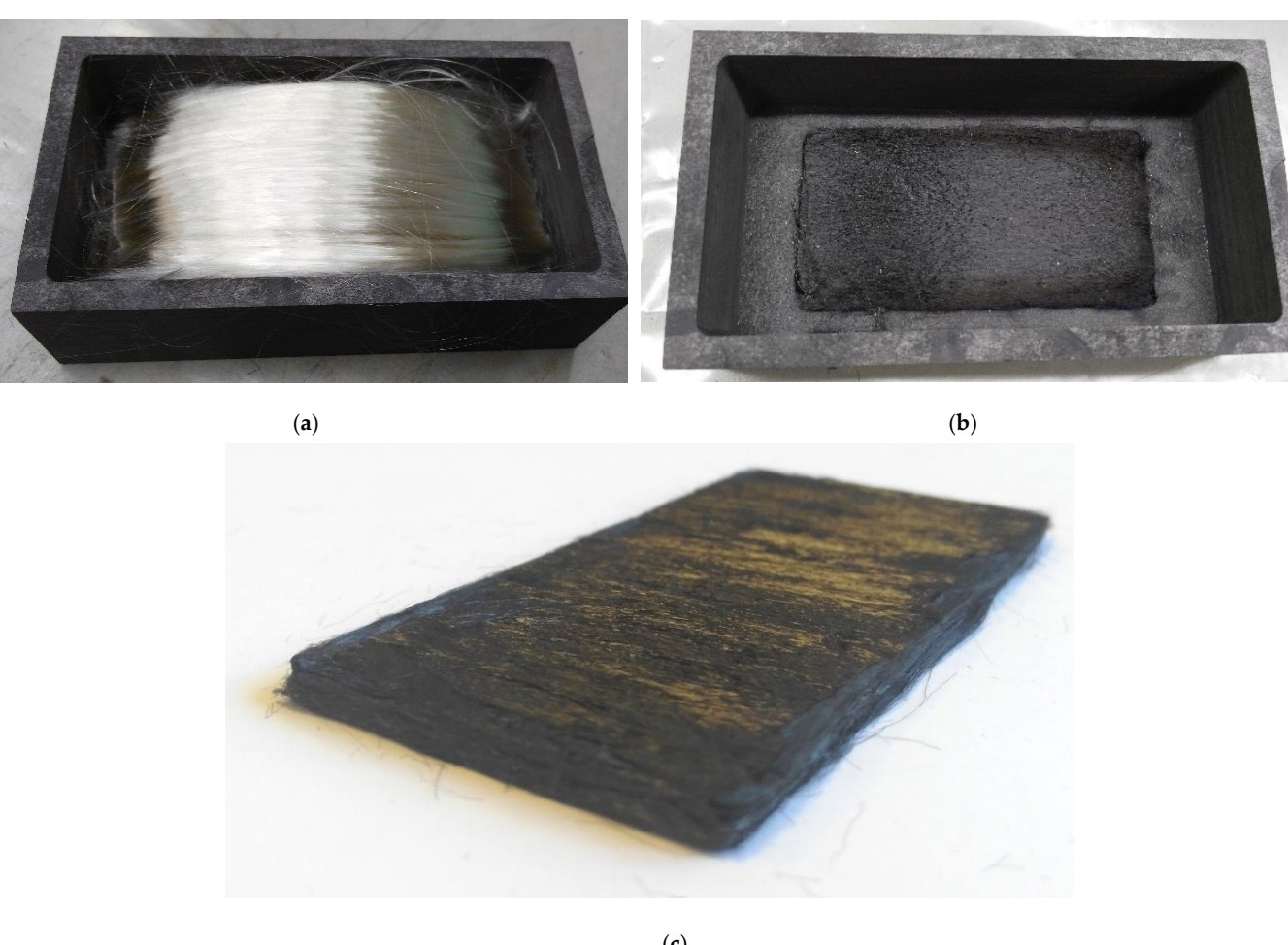

(**a**)  (**b**)

(**c**)

**Figure 12.** OP-SiC: (**a**) unidirectional PCS tape in a container before pyrolysis; (**b**) SiC frame, 90 × 45 × 4 mm in size, reinforcement scheme 0°/0° [1]; and (**c**) SiC frame, 90 × 45 × 3.5 mm in size, reinforcement scheme 0°/90°.

Characteristics of the organomorphic silicon carbide preforms are given in Table 1.

**Table 1.** Characteristics of the organomorphic silicon carbide preforms.

| Parameter | Value |
|:---:|:---:|
| Density, g/cm$^3$ | 0.9–1.0 |
| Open/closed porosity, % | 51–53/0 |
| Fiber volume content in the preform, not less than, % | 42 |
| Fiber diameter in the preform, µm | 15–17 |
| Shrinkage in length, % | 25–30 |
| Shrinkage by fiber diameter, % | 15–20 |
| Mass loss during pyrolysis, % | 19–20 |

The diffraction pattern obtained for the SiC frame after PCS preform pyrolysis shows two maxima: one of angular width 2θ of about 10° and the other of −20° (Figure 13).

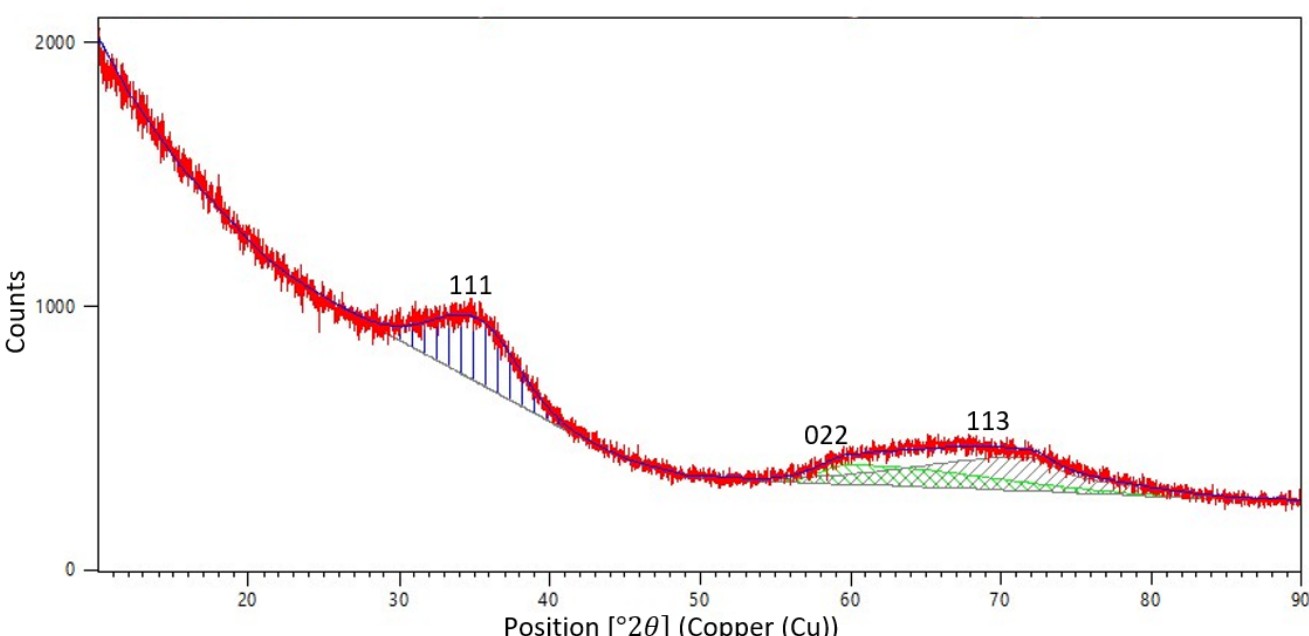

**Figure 13.** Diffraction pattern of the organomorphic SiC-preform (wide peak analysis was performed using the specialized HighScore Plus software for phase analysis).

Absence of narrow (angular width $2\theta < 2°$) diffraction peaks from the crystalline phase is indicative of a disordered—close to amorphous—state of the frame fibers. This is quite understandable, since the structure of the initial PCS fibers is completely amorphous (Figures 2 and 3) and the maximum temperature for obtaining the organomorphic SiC framework (1250 °C) does not reach even half of the incongruent decomposition temperature of silicon carbide [12]. However, the first—less wide—maximum corresponds to the strongest line (111) of β-SiC (reference code 98-002-8389), since this modification is formed during annealing of silicon carbide in a nitrogen atmosphere [13]. The second wide peak is apparently a superposition of other two strong lines, namely, (022) and (113).

Analysis of the silicon carbide preform microstructure confirms presence of some sort of bridges binding the fibers to each other (Figure 14).

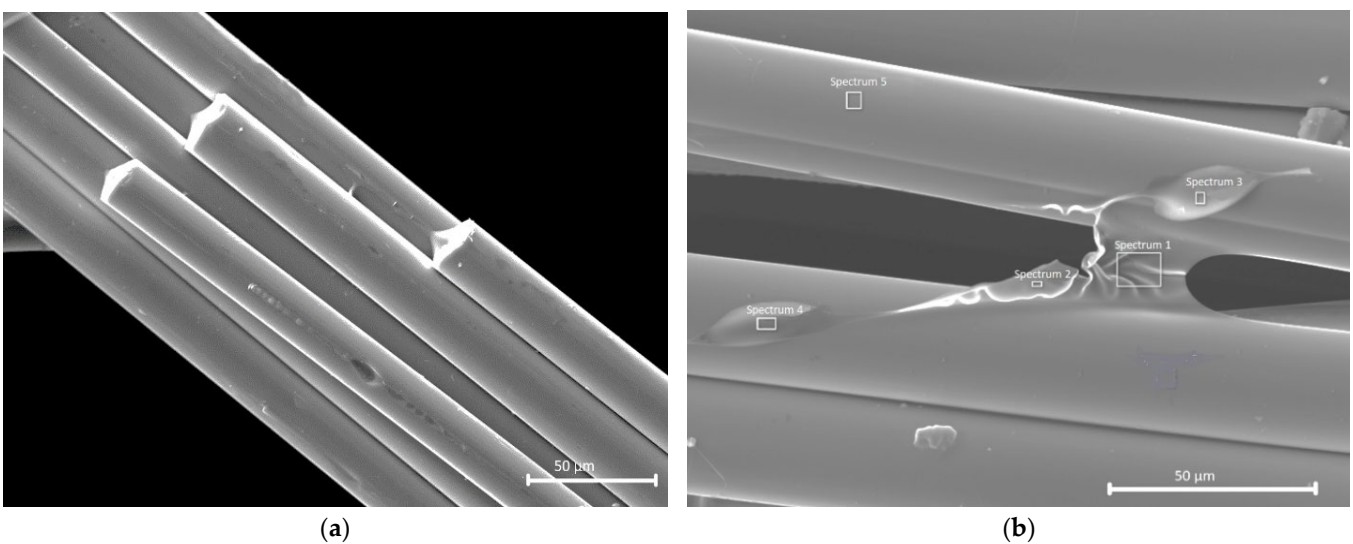

(**a**) (**b**)

**Figure 14.** SiC frame microstructure showing filament self-bonding areas (**a**,**b**). The boxes (spectrum 1 to 5) correspond to the areas for which the energy dispersive spectroscopy results were obtained.

According to energy dispersive spectroscopy results (Table 2), the chemical composition of the binding bridge (see Figure 14b; spectrum 1 and spectrum 2) can be represented approximately as Si:C:O = 1:13:1, which indicates enrichment with carbon.

**Table 2.** Energy dispersive spectroscopy results of the SiC-filaments within the preform after pyrolysis at 1250 °C.

| Element | Content, % at. | | | | |
|---|---|---|---|---|---|
| | **Spectrum 1** | **Spectrum 2** | **Spectrum 3** | **Spectrum 4** | **Spectrum 5** |
| Si | 6.45 | 6.41 | 23.42 | 33.25 | 26.98 |
| C | 84.81 | 86.71 | 55.22 | 48.47 | 54.51 |
| O | 8.74 | 6.88 | 21.36 | 18.28 | 18.51 |

The carbon content decreases sharply in the buildups (spectrum 3, spectrum 4) and their composition practically corresponds to the composition of the surface of the fiber itself (spectrum 5). Since the morphology of the SiC fiber buildup is similar to the buildup's nature of the PCS fibers shown in Figure 8, one can suggest similarity of their origin. The excessive carbon in spectrum 1 and spectrum 2 suggests that the carbon polymer can prevail over the organosilicon one in the buildups and accretions. This reveals possible composition difference among the filaments coming out on the surface to form buildups that are composed of parts of PCS macromolecules in the preform under pressing during the low-temperature (up to 400 °C) part of pyrolysis.

## 4. Discussion

The results obtained place a new light on the processes occurring during PCS fiber pyrolysis. Annealing of loose fibers results in production of also loose silicon carbide fibers. Pyrolysis of the PCS fiber wound on a spool results in poor unwindability of the resulting SiC fibers, as during carbidization, the PCS fibers when shrinking, compress the underlying layers. Finally, PCS fiber pyrolysis under load produces a dense, highly processible silicon carbide framework. Thus, one should recognize that the main factor affecting self-bonding of the PCS fibers during organic-to-inorganic state transition is their compression while being heated. Gluing of the fibers that not only prevents their unwinding during pyrolysis in the wound state but also contributes to organomorphic frame cementation is the result of the formation of buildups or accretions on the fiber surface from the matter released from the PCS fibers during the low-temperature part of pyrolysis under compressive force.

According to [14], for the half-width of the projection of elastic deformation area $r_o$ of two equal parallel cylinders of the same material (e.g., PCS fibers in a unidirectional preform), we have:

$$r_o = 1.52(qR/2E)^{1/2} \tag{1}$$

where q—specific load per unit fiber length in a unidirectional organomorphic frame; R—PCS fiber radius; and E—PCS fiber elasticity modulus.

During contact, for the maximum stress that develops along the axis of the contact area, we have:

$$\sigma_{max} = 0.418(2qE/R) \tag{2}$$

The maximum tangential stress $\tau(0, z)$ falls on the following depth:

$$z_o/r_o = 0.8 \tag{3}$$

and makes up about one third of $\sigma_{max}$.

Estimates made using Formulas (1)–(3) show that with the $r_o$ value of about 4.5 μm, the maximum stress $\sigma_{max}$ values make up approximately 0.03 MPa and the maximum tangential stress of about 0.01 MPa develops at a depth of 3.6 μm. In addition, it should be

borne in mind that due to an uneven degree of the fiber contact as observed in the actual preforms, the stress values in some contact bundles may be significantly higher.

Thus, both at the contact area and at the depth comparable to the half-width of the contact area, well-marked normal and tangential stresses develop that may affect the state of PCS macromolecules in the polymer fibers during heating and its accompanying shrinkage.

As evidenced by the configuration of the bridge between the filaments (see Figure 14), deformation of the buildups may appear even in the form of a viscous flow. Moreover, the polymer coming out on the fiber surface may be capable of curing, which may cause certain changes in the nature and a sharp increase in the elasticity modulus at above 200 °C (see Figure 4). The other reason for modulus increase is the increase in the degree of fiber contact during TMA [15].

It is obvious that these buildups come out on the filament surface from the bulk and correlate with the fiber composition. Since thermal stabilization of the PCS fibers in air is of a diffusive nature, the parts of PCS macromolecules possessed of greater mobility may diffuse through the defects and discontinuities in the more cross-linked surface layer under stress-strain conditions and upon reaching the onset temperature of partial decomposition of oxygen crosslinks. Undoubtedly, they are capable of active autohesive interaction with the surface of adjacent filaments.

Variations in the buildup composition over a wide range of carbon, oxygen, and silicon content, as well as their different concentration on the PCS filament surface, are likely to be associated with different defects in the local areas of the fiber surface and with difference in the degree of stitching of their respective microvolumes. This makes different parts of the PCS macromolecules come out on the surface under stress-strain conditions. Another possible reason for formation of SiC frames is the release of various methylsilanes from the polymer fibers during pyrolysis of the PCS preforms [16], among which dimethylsilane $(CH_3)_2SiH_2$ was detected by mass spectroscopy. Decomposition of this compound accompanied by deposition of silicon carbide starts as early as 450–500 °C.

## 5. Conclusions

The carbon organomorphic preform formation mechanism studied earlier using PAN fibers as an example [2] showed that self-bonding of the organic fibers within the preform is due to the presence of a solid polymer film on the fiber surface and a continuous physical contact between the filaments during pyrolysis when combined with hot pressing.

The investigation into the organomorphic silicon carbide preform formation mechanism reveals similarity of its underlying laws with those inherent in the formation of carbon frames. However, the insular nature of the buildups composed of the matter released from the PCS fibers during the low-temperature (up to 400 °C) part of pyrolysis reduces the volume-wise continuity of self-bonding in the organomorphic SiC preform. All the while, the relative densities of the silicon carbide preforms are still higher than 40%.

Apparently, the fundamental similarity of the self-bonding nature of the PAN- and PCS-preforms allows us to formulate the following basic prerequisites for origination of the contacts between the fibers within the preforms:

1. Stabilized structure of the polymer fibers to exclude their melting during pyrolysis.

2. Chemical purity of the surface of the contacting polymer filaments (absence of any obstacles for autohesion, i.e., diffusive merging of the polymer fibers).

3. Sufficient continuous mechanical pressing of the fibers against each other during pyrolysis (a constant load).

4. Duration of the contact between the filaments, when in the polymer state, sufficient for the autohesion interaction to form.

However, the presumable deposition of a nanoscale SiC layer from dimethylsilane $(CH_3)_2SiH_2$ on the fiber surface may complicate, at a later stage, the task of obtaining high-strength SiC–SiC CMCs based on the silicon carbide organomorphic reinforcing framework.

**Funding:** This research received no external funding.

**Institutional Review Board Statement:** Not applicable.

**Informed Consent Statement:** Not applicable.

**Data Availability Statement:** All the data supporting the reported results can be found in this manuscript. No additional data are available in the publicly archived datasets.

**Acknowledgments:** The author extends his great appreciation to the colleagues who assisted in carrying out the research: I.O. Leipunsky, Head of the Laboratory of Nano- and Microstructural Materials Science at the Institute of Energy Problems of Chemical Physics of the Russian Academy of Sciences, D.V. Onuchin, Deputy Vice-Rector for Science at the Mendeleev University of Chemical Technology and D.M. Kiselkov, Senior Scientist at the Institute of Technical Chemistry UB of the Russian Academy of Sciences. All individuals included in this section have consented to the acknowledgement.

**Conflicts of Interest:** The author declares no conflict of interest.

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
