# Peer review of "Organomorphic Silicon Carbide Reinforcing Preform Formation Mechanism"

_jcs, doi:10.3390/jcs7020081_

Round 1
Reviewer 1 Report
1. Figure 6 does not indicate what (a) and (b) specifically refer to.
2. FIG. 4 and Lines 129-131 illustrate the increase of elastic modulus and the decrease of sample thickness during the heating process. It is not specified why the elastic modulus increases correspondingly in this process.
3. Lines 157-159 mentioned the difference in fiber surface morphology before and after the TMA test, but only the SEM results of samples after the TMA test were given, without the SEM results before the TMA test as a comparison, which was not convincing enough.
4. Figure 9 shows the difference between the compression recovery deformation of PCS and PAN fibers in the TMA test, indicating that more than one raw material was used in the experiment. However, PCS fibers were only introduced in line 59, without the description of PAN fibers.
5. It was mentioned in line 162-165 that the reason for the island bulge on the surface of PCS fiber is that compared with PAN fiber, PCS fiber is less likely to adhere to each other after compression, so it is easier to recover the original shape. Firstly, the SEM results of PCS fibers were only presented in this paper, without the SEM results of PAN fibers as a comparison. Secondly, the causal relationship between the two can be explained in more detail.
6. Figure 4 is not standard, it has both Russian and English, and the figure is not straight.
7. There seems to be a problem with 2θ of the XRD result (line 186,189), or is there another way of expression?
Reviewer 2 Report
This work investigates the initial fibers of polycarbosilane process of organomorphic silicon carbide preforms, and the analysis of silicon carbide preforms leads to the factors of SiC framework self-bonding. The manuscript is suggested to accepted after minor modification. Some comments:
1. The article format needs to be checked carefully, temperature units, density values should be followed by units, abbreviations, etc.
2. should analyze in detail the reasons for weight loss at each stage of PCS fibers.
3. What caused the change after 600 oC in Figure 6(a) and what is the graph in 6(b)?
4. Figure 8 should improve the clarity of the picture.
5. The absorption peak of 1440 is marked in Figure 9 and reference is added, and the absorption peak of 2110 is unobvious in TMA.
6. Fig. 12 Two broad peaks should be explained by references
Round 2
Reviewer 1 Report
The authors addressed most of my concerns
Reviewer 2 Report
This work is acceptable for publication.